

**Comparison of Global Observations and Trends of Total Precipitable Water Derived**

**from Microwave Radiometers and COSMIC Radio Occultation from 2006 to 2013**

Shu-peng Ho[1], Liang Peng[1], Carl Mears[2], Richard A. Anthes[1]

[1] University Corporation for Atmospheric Research, P.O. Box 3000, Boulder, CO. 80307-3000

[2]Remote Sensing Systems, Santa Rosa, California, USA,

Corresponding author address: Dr. Shu-Peng Ho, COSMIC Project Office, University

Corporation for Atmospheric Research, P. O. Box 3000, Boulder CO. 80307-3000

E-mail: spho@ucar.edu

17

20    Shu-Peng Ho, COSMIC Project Office, Univ. Corp. for Atmospheric Research, P. O.

21    Box 3000, Boulder CO. 80307-3000, USA (spho@ucar.edu)



**Abstract**
We compare atmospheric total precipitable water (TPW) derived from SSM/I (Special
Sensor Microwave Imager) and SSMIS (Special Sensor Microwave Imager Sounder)
radiometers and WindSat to collocated TPW estimates derived from COSMIC
(Constellation System for Meteorology, Ionosphere and Climate) radio occultation (RO)
under clear and cloudy conditions over the oceans from June 2006 to December 2013.
Results show that the mean microwave (MW) radiometer - COSMIC TPW differences
range from 0.06-0.18 mm for clear skies, 0.79-0.96 mm for cloudy skies, 0.46-0.49 mm for
cloudy but non-precipitation conditions, and 1.64-1.88 mm for precipitation conditions.
Because RO measurements are not significantly affected by clouds and precipitation, the
biases mainly result from MW retrieval uncertainties under cloudy and precipitating
conditions. All COSMIC and MW radiometers detect a positive TPW trend over these eight
years. The trend using all COSMIC observations collocated with MW pixels is 1.79
mm/decade, with a 95% confidence interval of (0.96, 2.63), which is in close agreement
with the trend estimated by all MW observations (1.78 mm/decade with a 95% confidence
interval of 0.94, 2.62). These two trends from independent observations are larger than
previous estimates and are a strong indication of the positive water vapor-temperature
feedback in a warming planet.






## 1. Introduction


Clouds are important regulators for Earth's radiation and hydrological balances.
Water vapor is a primary variable that affects cloud radiative effects and hydrological
feedbacks. Accurate observations of long-term water vapor under both clear and cloudy
skies are important for understanding the role of water vapor on climate, which is still one
of the largest uncertainties in understanding climate change mechanisms (IPCC 2013).
Trends in global and regional vertically integrated total atmospheric water vapor, or Total
Precipitable Water (TPW), are important indicators of climate warming because of the
strong positive feedback between temperature and water vapor increases. Accurate
observations of TPW are therefore important in identifying climate change and in verifying
climate models, which estimate a wide range of TPW trends (Roman et al. 2014).
The TPW depends on temperature (Trenberth and Guillemot, 1998; Trenberth et
al., 2005). Global TPW can be derived from satellite visible, infrared, and microwave
sensors (i.e., Wentz and Spencer, 1998; Fetzer et al. 2006; John and Soden, 2007; Fetzer
et al. 2008; Noël et al. 2004). However, no single remote sensing technique is capable of
completely fulfilling the needs for climate studies in terms of spatial and temporal
coverage and accuracy. For example, while water vapor retrievals from visible and
infrared satellite sensors are limited to clear skies over both lands and oceans, passive
microwave (MW) imagers on satellites can provide all sky water vapor products, but only
over oceans. These water vapor products are mainly verified by comparing to either
reanalysis, radiosonde measurements, or other satellite data (i.e., Soden, and Lanzante,
1996; Sohn and Smith, 2003; Noël et al. 2004; Palm et al. 2008; Sohn and Bennartz, 2008;
Wick et al. 2008; Milz et al. 2009; Prasad and Singh, 2009; Pougatchev et al. 2009;



Knuteson et al., 2010; Larar et al. 2010; Wang et al. 2010; Ho et al. 2010a, b). Results from
these validation studies show that the quality of water vapor data from different satellite
sensors varies under different atmospheric conditions. The change of reanalysis systems
and inconsistent calibration among data may also cause uncertainty in long-term stability
of water vapor estimates. In addition, it is well known that radiosonde sensor characteristics
can be affected by the changing environment (Luers and Eskridge, 1998; Wang and Zhang,
2008). Ho et al. (2010b) demonstrated that the quality of radiosonde humidity
measurements varies with sensor types, adding extra difficulties in making a consistent
validation of long term water vapor products.
MW imagers are among the very few satellite instruments that are able to provide
long-term (close to 30 years) all-weather time series of water vapor measurements using
similar sensors and retrieval techniques (Wentz, 2015). The measured radiances at 19.35,
22.235, and 37.0 GHz from SSMIS and 18.7, 23.8, and 37.0 GHz from WindSat are used
to derive TPW, total cloud water (TCW), wind speed, and rainfall rates over oceans (Wentz
and Spencer, 1998). These four variables are retrieved by varying their values until the
brightness temperatures calculated using a forward model match satellite-observed
brightness temperatures. Because MW radiation is significantly affected (absorbed or
scattered) by heavy rain, these four variables are only retrieved under conditions of no or
light-to-moderate rain (Schlüssel and Emery, 1990; Elsaesser and Kummerow, 2008;
Wentz and Spencer, 1998).
Recently version 7.0 daily ocean products mapped to a 0.25° grid derived from
multiple MW radiometers were released by Remote Sensing System (RSS) (Wentz, 2013).
Many validation studies have been performed by RSS by comparing the MW TPW





retrievals with those from ground-based Global Positioning System (gb-GPS) stations
(Mears et al, 2015; Wentz, 2015). Because the gb-GPS stations are nearly always located
on land, these validation studies use stations located on small and isolated islands (Mears
et al., 2015). RSS results for TPW collocated with those derived from gb-GPS over these
island stations show that their mean differences vary from station to station, and can be as
large as 2 mm. The mean difference also varies with surface wind speed, varying from 1
mm at low wind speeds to -1 mm at high wind (20 m/s) speeds. The difference is near zero
for the most common wind speeds (6 to 12 m/s). Because the uncertainty of the input
parameters and change of antenna for each GPS receiver (Bock et al., 2013), the mean
TPW(RSS) – TPW (gb-GPS) can vary from -1.5 mm to 1.5 mm for a single MW radiometer
(see Figure 4 in Mears et al., 2015). Wentz (2015) compared 17 years of Tropical Rainfall
Measuring Mission (TRMM) Microwave Imager (TMI) TPW collocated with gb-GPS
TPW over the region from 45°N to 45° S. The mean TMI- gb-GPS TPW bias was estimated
to be 0.45 mm with a standard deviation ($\sigma$) of 2.01 mm.

Unlike passive MW radiometers and infrared sensors, radio occultation (RO) is an

active remote sensing technique. RO can provide all-weather, high vertical resolution (from
~100 m near the surface to ~1.5 km at 40 km) refractivity profiles (Anthes, 2011). The
basics of the RO measurement is a timing measured against reference clocks on the ground,
which are timed and calibrated by the atomic clocks at the National Institutes for Standards
and Technology (NIST). With a GPS receiver onboard the LEO (Low-Earth Orbiting)
satellite, this technique is able to detect the bending of radio signals emitted by GPS
satellites and traversing the atmosphere. With the information about the relative motion of
the GPS and LEO satellites, the bending angle profile of the radio waves can be used to



derive all-weather refractivity, pressure, temperature, and water vapor profiles in the
neutral atmosphere (Anthes et al., 2008).

Launched in June 2006, COSMIC (Constellation Observing System for

Meteorology, Ionosphere, and Climate) RO data have been used to study atmospheric
temperature and refractivity trends in the lower stratosphere (Ho et al., 2009a, b, and 2012),
and modes of variability above, within, and below clouds (Biondi et al., 2012, 2013; Teng
et al., 2013; Scherllin-Pirscher et al., 2012; Zeng et al., 2012; Mears et al., 2012). Wick et
al., (2008) (Wick2008 hereafter) demonstrated the feasibility of using COSMIC-derived
TPW to validate SSM/I TPW products over the east Pacific Ocean using one month of
data. Many studies have demonstrated the usefulness of RO derived water vapor to detect
climate signals of El Niño–Southern Oscillation (ENSO; Teng et al., 2013; Scherllin-
Pirscher et al, 2012; Huang et al., 2013), Madden-Julian Oscillation (MJO; Zeng et al.,
2012), and improving moisture analysis of atmospheric rivers (Neiman et al., 2008; Ma et
al. 2011).

The objective of this study is to use COSMIC RO TPW to characterize the global

TPW values and trends derived from multiple MW radiometers over oceans, including
under cloudy and precipitating skies. COSMIC TPW from June 2006 to December 2013
are compared to the co-located TPW derived from MW radiometers over the same time
period. Because RO data are not strongly sensitive to clouds and precipitation, COSMIC
TPW estimates can be used to identify possible MW TPW biases under different
meteorological conditions. We describe datasets and analysis method used in the
comparisons in Section 2. The comparison results under clear skies and cloudy skies are



summarized in Sections 3 and 4, respectively. The time series analysis is in Section 5. We
conclude this study in Section 6.

**2. RSS Version 7.0 Data and COSMIC TPW Data and Comparison Method**
**2.1 RSS Version 7.0 Data Ocean Products**
The RSS version 7.0 ocean products are available for SSM/I, SSMIS, AMSR-E,
WindSat, and TMI. The inversion algorithm is mainly based on Wentz and Spencer,
(1998), where above a cutoff in liquid water column, water vapor is no longer
retrieved. The various radiometers from the different satellites have been precisely inter-
calibrated at the radiance level by analyzing the measurements made by pairs of satellites
operated at the same time. This was done for the explicit purpose of producing versions of
the datasets that can be used to study decadal-scale changes in TPW, wind, clouds, and
precipitation, so special attention was focused on inter-annual variability in instrument
calibration. The calibration procedures and physical inversion algorithm used to
simultaneously retrieve TPW, surface wind speed (and thereby surface wind stress and
surface roughness) and the total liquid water content are summarized in Wentz (2013) and
Wentz (1997), respectively. This allows the algorithm to minimize the effect of wind speed,
clouds, and rain on the TPW measurement.
The RSS version 7.0 daily data are available on a 0.25°x0.25° grid for daytime and
nighttime (i.e., 1440x720x2 daily per day). Figures 1a-d shows the RSS V7.0 monthly
mean F16 SSMIS TPW (in mm), surface skin temperature (in K), liquid water path (LWP,
in mm), and rain rate (RR, in mm/hour), respectively. Figure 1 shows that the variation and
distribution of TPW over oceans (Fig. 1a) is, in general, closely linked to surface skin





temperature (Fig. 1b), which is modulated by clouds and the hydrological cycle (Soden et
al., 2002). The distribution of monthly TPW is consistent with those of cloud water vapor
distribution patterns where highest TPW values (and LWP and RR) occur in persistent
cloudy and strong convective regions over the tropical west Pacific Ocean near Indonesia.
Because COSMIC reprocessed TPW data are only available from June 2006 to
December 2013 (i.e., COSMIC2013), the SSM/I F15, SSMIS F16, SSMIS F17, together
with WindSAT RSS Version 7.01 ocean products covering the same time period are used
in this study. Table 1 summarizes the starting date and end date for RSS SSM/I F15, SSMIS
F16, SSMIS F17, and WindSAT data. The all sky daily RSS ocean products for F15, F16,
F17, and WindSat are downloaded from http://www.remss.com/missions/ssmi.

**2.2 COSMIC TPW Products**

The atmospheric refractivity N is a function of the pressure P, temperature T, water
vapor pressure $P_W$, and water content W through the following relationship (Kursinski
1997; Zou et al. 2012):

$$N = 77.6\frac{P}{T} + 3.73 \times 10^5 \frac{P_W}{T^2} + 1.4W_{water} + 0.61W_{ice} \qquad (1)$$


where P is pressure in hPa, T is temperature in K, $P_W$ is water vapor pressure in hPa,
$W_{water}$ is liquid water content in grams per cubic meter ($gm^{-3}$), and $W_{ice}$ is the ice water
content in $gm^{-3}$. The last two terms generally contribute less than 1% to the refractivity and
may be ignored (Zou et al., 2012). However, they can be significant for some applications
under conditions of high cloud liquid or ice water content, as shown by Lin et al. 2010;





Yang and Zou 2012; Zou et al. 2012. We will neglect these terms in this study, but because
we are looking at small differences between MW and RO TPW in cloudy and precipitating
conditions in this paper, we estimate the possible contribution of these terms to RO TPW
and the consequences of neglecting them here. Since both of these terms increase N,
neglecting them in an atmosphere in which they are present will produce a small positive
bias in water vapor pressure $P_W$ and therefore total precipitable water when integrated
throughout the entire depth of the atmosphere.

Typical value of cloud LWC range from ~0.2 $gm^{-3}$ in stratiform clouds (Thompson,

2007) to 1 $gm^{-3}$ in convective clouds (Thompson, 2007; Cober et al. 2001). Extreme values
may reach ~2 $gm^{-3}$ in deep tropical convective clouds (i.e., cumulonimbus). Ice water
content values are smaller, typically 0.01 – 0.03 $gm^{-3}$ (Thompson, 2007).

For extremely high values of $W_{water}$ and $W_{ice}$ of 2.0 and 0.5 $gm^{-3}$, the contributions

to N are 2.8 and 0.3 respectively. The values of N in the atmosphere decrease exponentially
upward, from ~300 near the surface to ~150 at P=500 hPa.  Using the above extreme values
at 500 hPa, $W_{water}$ may contribute from up to 1.6% of N and $W_{ice}$ up to 0.2%. Thus we may
assume that in most cases the error in N due to neglecting these terms will be less than 1%.
The effect on TPW will be even less, since clouds do not generally extend through the full
depth of the atmosphere. Finally, the ~200 km horizontal averaging scale of the RO
observation footprint makes it unlikely that such extremely high values of water and ice
content will be present over this scale. We conclude that the small positive bias in RO TPW
introduced by neglecting the liquid and water terms in (1) will be less than 1%.

To resolve the ambiguity of COSMIC refractivity associated with both temperature

and  water  vapor  in  the  lower  troposphere  a  1D-var  algorithm  (http://cosmic-



io.cosmic.ucar.edu/cdaac/doc/documents/1dvar.pdf) is used to derive optimal temperature
and water vapor profiles while temperatures and water vapor profiles from the ERA-
Interim reanalysis are used as a priori estimates (Neiman et al. 2008; Zeng et al. 2012).

Note that because RO refractivity is very sensitive to water vapor variations in the

troposphere (Ho et al. 2007), and is less sensitive to temperature errors, RO-derived water
vapor product is of high accuracy (Ho et al. 2010 a, b). It is estimated that 1K of
temperature error will introduce less than 0.25 g/kg of water vapor bias in the troposphere
in the 1D-var retrievals. Although the first guess temperature and moisture are needed for
the 1D-Var algorithm, the retrieved water vapor profiles are insensitive to the first guess
water vapor profiles (Neiman et al. 2008).

The horizontal footprint of a COSMIC observation is about 200 km in the lower

troposphere and its vertical resolution is about 100 m near the surface and 1.5 km at 40 km.
The COSMIC post-processed water vapor profiles version 2013.2640 collected from
COSMIC Data Analysis and Archive Center (CDAAC)
(http://cosmicio.cosmic.ucar.edu/cdaac/index.html) are used to construct the COSMIC
TPW data. To further validate the accuracy of COSMIC-derived water vapor, we have
compared COSMIC TPW with those derived from ground-based GPS (i.e., International
Global Navigation Satellite Systems–IGS, Wang et al. 2007) which are assumed to be
independent of location. Only those COSMIC profiles whose lowest penetration heights
are within 200 meters of the height of ground-based GPS stations are included. Results
showed that the mean global difference between IGS and COSMIC TPW is about -0.2 mm
with a standard deviation of 2.7 mm (Ho et al., 2010a). Similar comparisons were found
by Teng et al. (2013) and Huang et al. (2013).



### 2.3 Preparation of COSMIC TPW data for Comparison


In this study, only those COSMIC water vapor profiles penetrating lower than 0.1
km are integrated to compute TPW. To compensate for the water vapor amount below the
penetration height, we follow the following procedure:
i)      we assume the relative humidity below the penetration height is equal to 80%. This is

a good assumption especially over oceans near the sea surface (Mears et al., 2015);

ii)     the temperatures below the penetration height are taken from the ERA-interim

reanalysis;

iii)    we compute the water vapor mixing ratio below the penetration heights;
iv)     we integrate the TPW using COSMIC water vapor profiles above the penetration

heights with those water vapor profile below the penetration heights.

Pairs of MW and RO TPW estimates collocated within 50 km and one hour are collected.
Wick2008 used MW-RO pairs within 25 km and one hour in time. To evaluate the effect
of the spatial difference on the TPW difference, we also computed TPW differences for
MW-RO pairs within 75 km, 100 km, and 150 km, and 200 km. We found the larger spatial
difference increases the mean TPW biases slightly to +/- 0.25 mm and the standard
deviations to +/- 1.91 mm, which is likely because of the high spatial variability of water
vapor.
With a 0.25°×0.25° grid, there are about 20 to 60 MW pixels matching one
COSMIC observation. The number of pixels varies at different latitudes. A clear MW-RO
pair is defined as instances when *all* the TCW values for the collocated MW pixels are
equal to zero. A cloudy MW-RO ensemble is defined as instances when *all* the TCW values
from the collocated MW pixels are larger than zero. Partly cloudy conditions (some of



pixels zero and some non-zero) are excluded from this study. The cloudy ensembles are
further divided into precipitating and non-precipitating conditions. MW-RO pairs are
defined as cloudy non-precipitating when less than 20% of MW pixels have rainfall rates
larger than zero mm/hour. Cloudy precipitating MW-RO pairs are defined when more than
20% of the pixels have rainfall rates larger than zero. Because microwave radiances are not
sensitive to ice, we treat cloudy pixels of low density like cirrus clouds as clear pixels.

**3. Comparison of MW and RO TPW with clear skies**

In total there are 26,678 F15-RO pairs, 31,610 F16-RO pairs, 31,291 F17-RO pairs,

and 21,996 WindSat-RO pairs from June 2006 to Dec. 2013, respectively. Because
microwave radiances are not sensitive to ice, we treat cloudy pixels of low density like
cirrus clouds as clear pixels. Figures 2a-d show scatter plots for F15-COSMIC TPW, F16-
COSMIC TPW, F17-COSMIC TPW, and WindSat-COSMIC TPW under clear skies.
Figures 3a-d show that MW clear sky TPW from F15, F16, F17, and WindSat are all very
consistent with those from co-located COSMIC observations. As summarized in Table 2,
under clear conditions where SSM/I provides high quality TPW estimates, the mean TPW
biases between F16 and COSMIC (F16- COSMIC) is equal to 0.03 mm with a standard
deviation σ of 1.47 mm. The mean TPW differences are equal to 0.06 mm with a σ of 1.65
mm for F15, 0.07 mm with a σ of 1.47 mm for F17, and 0.18 mm with a σ of 1.35 mm for
WindSat. The reason for larger standard deviation for F15 may be because the F15 data
after August 2006 were corrupted by the "rad-cal" beacon that was turned on at this time
(Hilburn and Wentz, 2008). F16 had solar radiation intrusion into the hot load during the
time period, while F17 and WindSat had no serious issues.




## 4. Global comparisons of MW and RO TPW with cloudy skies

### 4.1 Comparison of MW, RO, and Ground-based GPS TPW



Figures 3a-c depict the scatter plots for F16-COSMIC pairs under cloudy, cloudy

non-precipitation, and precipitation conditions from June 2006 to December 2013 over
oceans. While there is a very small bias (0.031 mm) for clear pixels (Fig. 2b), there is a
significant positive TPW bias (0.794 mm) under cloudy conditions (Fig. 3a). This may
explain the close to 0.4 mm mean TMI-gb GPS TPW biases found by Wentz et al., (2015)
where a close to 7 years of data were used. Figure 3c depicts that the large SSM/I TPW
biases under cloudy skies are mainly from the pixels with precipitation (mean bias is equal
to 1.825 mm) although precipitation pixels are of about less than 6% of the total F16–
COSMIC pairs. Because RO measurements are not significantly affected by clouds and
precipitation, the biases mainly result from MW retrieval uncertainty under cloudy
conditions. The fact that the MW-COSMIC biases for precipitation conditions (1.825 mm,
Fig. 3c and 1.62-1.88 mm in Table 2) is much larger than those for cloudy, but non-
precipitation conditions, indicates that significant scattering and absorbing effects are
present in the passive MW measurements when it rains. The correlation coefficients for
F15-RO, F16-RO, F17-RO, and WindSat-RO pairs for all sky conditions are all larger than
0.96 (not shown).

297       MW and gb-GPS TPW comparisons show similar differences as the MW-RO

differences under the different sky conditions. We compared F16 pixels with those from
gb-GPS within 50 km and 1 hour over the 33 stations studied by Mears et al. (2015) from
2002 to 2013. Figs. 4a-d depicts the scatter plots for F16-gb-GPS TPW under clear, cloudy,



cloudy non-precipitating, and cloudy precipitating conditions, respectively. The F16-gb-
GPS mean biases are equal to 0.241 mm (clear skies), 0.614 mm (cloudy skies), 0.543 mm
(cloudy-non precipitation) and 1.197 mm (precipitation), which are similar to those
estimated from MW-RO comparisons (Table 2).

The above results show that the MW estimates of TPW are biased positively

compared to both the RO and the ground-based GPS estimates, which are independent
measurements. The biases are smallest for clear skies and largest for precipitating
conditions, with cloudy, non-precipitating biases in between. Overall, the results suggest
that clouds and especially precipitation contaminate the MW radiometer measurements to
a small but significant degree.

**4.2 Time Series of MW, RO, and Ground-based TPW Biases under Various**
**Meteorological Conditions**

To further examine how rain and cloud drops affect the MW TPW retrievals, we

show how the F16-RO TPW biases vary under different meteorological conditions in Fig.
5. The bias dependence on wind speed (Fig. 5a) is small. Unlike the results from Mears et
al., (2015), the mean TPW biases between F16 and COSMIC are within 0.5 mm with high
winds (wind speed larger than 20 m/s). Fig. 5b indicates that the F16-COSMIC bias is
larger with TPW greater than about 10 mm, which usually occurs under cloudy conditions.
The F16-COSMIC biases can be as large as 2.0 mm when the rainfall rate is larger than 1
mm/hour (Fig. 5c), which usually occurs with high total liquid cloud water (Fig. 5d)
conditions. The F16 TPW biases can be as large as 2 mm when total cloud water is larger
than 0.3 mm (Fig. 5d). Fig. 5e shows that the larger F16-COSMIC TPW biases (2-3 mm)



mainly occur over regions with surface skin temperature less than 270 K (higher latitudes,
see Figure 1b). The F15, F17, and WindSat TPW biases under different meteorological
conditions are very similar to those of F16 (not shown).

In Figure 6 we compare RSS V7.0 F16 MW TPW to the ground-based GPS TPW

over various meteorological conditions. The magnitudes of the MW-gb-GPS TPW
differences under high rain rate and high total cloud water conditions are somewhat smaller
than those of MW-RO pairs (varying from about 0.5 mm to 2.0 mm), which may be because
most of the MW-gb-GPS samples are collected under low rain rates (less than 1 mm/hour)
conditions.

**5. Eight Year Time Series and Trend Analysis under All Skies**
**5.1 Monthly Mean TPW Time Series Comparison**

To further examine MW TPW long-term stability and trend uncertainty due to rain

and water drops for different instruments, we compared time series of the MW and
COSMIC monthly mean TPW differences from June 2006 to Dec. 2013. Figures 7a-d show
the monthly mean F16-COSMIC TPW differences from June 2006 to Dec. 2013 for clear,
cloudy, cloudy non-precipitation, and precipitation conditions. In general, the microwave
TPW biases under different atmospheric conditions are positive and stable from June 2006
to Dec. 2013, as reflected in relatively small standard deviation values (Table 3). Except
for F15, the standard deviations of the monthly mean TPW anomaly range are less than
0.38 mm (Table 3). In contrast, the F15-COSMIC monthly mean σ range from 0.48 mm to
0.69 mm with different conditions.



Table 3 also shows the trend in the RO estimates of TPW over the eight-year period

of study. The trends are range from -0.12 mm/decade (WindSat, clear skies) to 2.52
mm/decade (F15, precipitation conditions). The overall trend is positive as discussed in the
next section. Table 3 shows that in general the trends are more strongly positive under
cloudy and precipitation conditions compared to clear conditions.

**5.2 De-seasonalized Trends of MW-RO Differences and TPW**

Fig. 8 depicts the de-seasonalized trends of the MW-RO TPW differences under

cloudy skies for F15 (Fig. 8a), F16 (Fig. 8b), F17 (Fig. 8c), and Windsat (Fig.8d). Except
for F15, the de-seasonalized trends of the MW-RO TPW differences for the MW
radiometers are close to zero, indicating little change over these eight years. The trends of
the biases associated with F16, F17, F18 and WindSat under all sky conditions range from
-0.09 to 0.27 mm/decade (details not shown).

The reason for larger standard deviations of the MW minus RO differences for F15

(Tables 2 and 3 and Fig. 8a) is very likely because the F15 data after August 2006 were
corrupted by the "rad-cal" beacon that was turned on at this time. Adjustments were derived
and applied to reduce the effects of the beacon, but the final results still show excess noise
relative to uncorrupted measurements (Hilburn and Wentz, 2008). RSS does not
recommend using these measurements for studies of long-term change.  Thus we consider
the F15 data less reliable during the period of our study.

Fig. 9 shows the de-seasonalized time series of global monthly mean TPW for all

MW radiometers and COSMIC under all sky conditions. The close to eight year trends for
TPW estimated from both passive MW radiometers and active COSMIC RO sensors are



positive and very similar in magnitude. The global mean trend of COSMIC RO TPW is
1.79 mm/decade with a 95% confidence interval of (0.96, 2.63) mm/decade while the
global mean trend from all the MW estimates is 1.78 mm/decade with a 95% confidence
interval of (0.94, 2.62). This close agreement between completely independent
measurements lends credence to both estimates.

The positive trends in TPW from the independent MW and RO measurements

reflect the positive feedback between TPW and global warming and are considerably
greater than previous estimates over other time periods. For example, Durre et al. (2009)
estimated a trend of 0.45 mm/decade for the Northern Hemisphere over the period 1973-
2006. Trenberth et al. (2005) estimated a global trend of 0.40 +/- 0.09 mm/decade for the
period 1988 to 2001. The 100-year trend in global climate models is variable, ranging from
0.55 to 0.72 mm/decade (Roman et al., 2014).

Although both MW and RO measurements demonstrate positive trends in TPW

from 2006 to 2013, the trends in TPW vary over different regions. Fig. 10 shows the global
map of TPW trend over oceans using all F16, F17, and WindSat data from 2006 to 2013.
Fig. 10 shows that the positive trends in TPW exist mainly over central and north Pacific
oceans, south of China and west of Austrian, south of South American, and east of
America.

**6. Discussion and conclusions**

RSS water vapor products have been widely used for climate research. The newly

available RSS V7.0 data products have been processed using consistent calibration
procedures (Wentz, 2013). This was done for the explicit purpose of producing versions of



the datasets that can used to study decadal scale changes in TPW, wind, clouds, and
precipitation. These water vapor products are mainly verified by comparing to either
reanalysis, radiosondes measurements, or other satellite data. However, because the quality
of these datasets may also vary under different atmospheric conditions, the uncertainty in
long-term water vapor estimates may still be large. In this study, we used TPW estimates
derived from COSMIC active RO sensors to identify TPW uncertainties from four different
MW radiometers under clear, cloudy, cloudy/non-precipitation, and cloudy/precipitation
skies over nearly eight years (from June 2006 to Dec. 2013). Because RO data are not
sensitive to clouds and precipitation, RO-derived water vapor products are useful to
identify the possible TPW biases retrieved from measurements of passive microwave
imagers under different sky conditions. We reach the following conclusions:

1) Clear sky biases: The collocated COSMIC RO TPW estimates under clear skies

are highly consistent with the MW TPW estimates under clear sky conditions (within +/-
0.2 mm and with a correlation coefficient greater than 0.96). The mean TPW bias between
F16 and COSMIC (F16- COSMIC) is equal to 0.03 mm with a standard deviation $\sigma$ of 1.47
mm. The mean TPW differences are equal to 0.06 mm with a $\sigma$ of 1.65 mm for F15, 0.07
mm with a $\sigma$ of 1.47 mm for F17, and 0.18 mm with a $\sigma$ of 1.35 mm for WindSat. These
small values give us confidence in our approach to interpolate COSMIC data the last few
hundred meters below the lowest penetration height. The consistent F15-COSMIC, F16-
COSMIC, F17-COSMIC, and WindSat-COSMIC TPW under clear skies also show that
COSMIC TPW can be used as good references to identify and correct TPW among
different MW imagers for other sky conditions.





2) Biases under cloudy skies: while there are very small biases for clear pixels,
there are significant positive MW TPW biases (~0.80 mm) under cloudy conditions when
compared to RO TPW. The large SSM/I TPW biases under cloudy skies result mainly from
the pixels with precipitation. The mean bias is equal to 1.83 mm for COSMIC-F16 pairs,
which is much larger than the bias for cloudy, but non-precipitation conditions. This
indicates that the significant scattering and absorbing effects are present in the passive MW
measurements when it rains. The F16 – Ground-based GPS mean biases are equal to 0.24
mm (for clear skies), 0.61 mm (for cloudy skies), 0.54 mm (for cloudy-non precipitation)
and 1.2 mm (for precipitation) which are consistent with those from F16-COSMIC
comparisons.
3) Biases among different instruments: using RO TPW estimates collocated with
different MW instruments, we are able to identify possible TPW inconsistencies among
MW instruments even they are not collocated. The de-seasonalized trends in MW-RO TPW
differences for three MW radiometers (i.e., F16, F17, and WindSat) are close to zero,
indicating consistency among these radiometers. However, the F15-COSMIC differences
are larger and show a significant trend over the eight years of the study. It is likely that F15
data after August 2006 were corrupted by the "rad-cal" beacon that was turned on at this
time.
4) Trend of TPW under all skies: The eight-year trends of TPW estimated from
both passive MW radiometer and active COSMIC sensors show increasing TPW globally,
with higher trends under cloudy conditions. The global mean trend of COSMIC RO TPW
collocated with MW observations is 1.79 mm/decade with a 95% confidence interval of
(0.96, 2.63) mm/decade. The global mean trend from all the MW estimates is 1.78





mm/decade with a 95% confidence interval of (0.94, 2.62). This close agreement between
completely independent measurements lends credence to both estimates. These trends are
significantly higher than other estimates and are a strong confirmation of the water vapor-
temperature feedback in a warming global atmosphere.

**Acknowledgements.** This work is supported by the NSF CAS AGS-1033112. We thank
Eric DeWeaver (NSF) and Jack Kaye (NASA) for sponsoring this work.




















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












Table 1. Satellite Instruments Used in This Study

| Satellite | Instrument | Operation period |
|---|---|---|
| DMSP F15 | SSM/I | December 1999-present |
| DMSP F16 | SSMIS | October 2003-present |
| DMSP F17 | SSMIS | December 2006-present |
| Coriolis | WindSat | February 2003-present |

















Table 2: Mean and standard deviation of differences (MW minus RO) in TPW (in mm) between four MW radiometers and COSMIC RO under various sky conditions. The sample numbers for each pair are shown in the third position of each column.

| Sky condition | Mean/σ/N | | | |
| --- | --- | --- | --- | --- |
| | F15 | F16 | F17 | WindSat |
| Clear | 0.06/1.65/3064 | 0.03/1.47/3551 | 0.07/1.47/2888 | 0.18/1.35/1802 |
| Cloudy | 0.80/1.92/23614 | 0.79/1.73/29059 | 0.82/1.76/28403 | 0.96/1.73/20194 |
| Non Precip | 0.49/1.69/17223 | 0.46/1.46/21854 | 0.47/1.49/21371 | 0.49/1.36/13004 |
| Precip | 1.64/2.28/6391 | 1.83/2.05/7205 | 1.88/2.08/7032 | 1.85/2.00/7190 |





Table 3: Mean and standard deviation (std) of the mean in mm of the monthly time series
of differences of MW minus RO TPW under various sky conditions. The trend of the RO
estimates of TPW in mm/decade and the 95% confidence level are shown below the
mean and σ values in each row.

| Sky condition | Mean/σ of monthly time series RO trend (95% confidence levels indicated in ( ) | | | |
| --- | --- | --- | --- | --- |
| | F15 | F16 | F17 | WindSat |
| Clear | 0.07/0.56 | 0.05/0.28 | 0.08/0.27 | 0.23/0.38 |
| | 1.65 (0.47,2.84) | 1.09 (-0.28,2.46) | 0.21 (-1.22,1.65) | -0.12 (-1.89,1.66) |
| Cloudy | 0.77/0.51 | 0.78/0.18 | 0.82/0.15 | 0.95/0.17 |
| | 1.49 (0.40,2.58) | 2.02(0.87,3.16) | 1.85 (0.64,3.06) | 1.85 (0.68,3.01) |
| Non Precipitation | 0.46/0.48 | 0.45/0.17 | 0.48/0.15 | 0.47/0.19 |
| | 0.86 (-0.24,1.95) | 2.02 (0.87,3.17) | 2.37 (1.23,3.50) | 2.12 (0.95,3.30) |
| Precipitation | 1.62/0.69 | 1.81/0.31 | 1.88/0.29 | 1.88/0.32 |
| | 2.52 (0.55,4.480 | 1.32 (-0.53,3.17) | 0.26 (-1.59,2.10) | 0.39 (-1.25,2.04) |


















Figure Captions
Figure 1. The RSS V7.0 monthly mean F16 SSM/I a) TPW (in mm), b) surface skin
temperature (in K), c) liquid water path (LWP, in mm), and d) rain rate (RR, in
mm/hour).

Figure 2. TPW scatter plots for the COSMIC and RSS Version 7.0 pairs under clear
condition for a) F15, b) F16, c) F17, and d) WindSat.

Figure 3. TPW scatter plots for the COSMIC and RSS Version 7.0 F16 SSM/I pairs
under a) cloudy, b) cloudy but non-precipitation, and c) precipitation conditions.

Figure 4. TPW scatter plots for the gb-GPS and RSS Version 7.0 F16 SSM/I pairs from
June 2006 to December 2013 under a) clear, b) cloudy, c) cloudy but non-precipitation,
and d) precipitation conditions.

Figure 5. Mean and standard of the mean for the F16-COSMIC TPW biases varying with
a) wind speed (m/s), b) TPW (mm), c) rain rate (mm/hour), d) total cloud water (mm),
and e) surface skin temperature (K). The vertical black bracket superimposed on the
mean denotes the standard error of the mean. The green dashed line is the number of
samples, indicated by the scale on the right.




Figure 6. Mean and standard of the mean for the F16- gb-GPS TPW biases varying with
a) wind speed (m/s), b) TPW (mm), c) rain rate (mm/hour), d) total cloud water (mm) and
e) surface skin temperature (K).  The vertical black bracket superimposed on the mean
denotes the standard error of the mean. The green dashed line is the number of samples,
indicated by the scale on the right.

Figure 7. The time series of monthly mean F16 – COSMIC TPW differences under a)
clear, b) cloudy, c) cloudy but non-precipitation, and d) precipitation conditions. The
black line is the mean difference for microwave radiometer minus COSMIC; the vertical
lines superimposed on the mean values are the standard error of the mean. The number of
the monthly MW radiometer- COSMIC pairs is indicated by the green dashed line (scale
on the right Y axis).

Figure 8. The time series of de-seasonalized TPW differences (microwave radiometer –
COSMIC) under cloudy skies for a) F15, b) F16, c) F17, d) WindSat. The black line is
the mean difference for microwave radiometer minus COSMIC; the vertical lines
superimposed on the mean values are the standard error of the mean. The number of the
monthly MW radiometer- COSMIC pairs is indicated by the green dashed line (scale on
the right Y axis). The trends are shown by solid red line. The 95% confidence intervals
for slopes are shown in the parentheses.





Figure 9. The de-seasonalized time series of global monthly mean TPW for all MW
instruments and COSMIC under all sky conditions. The red and blue dashed lines are the
best fit of de-seasonalized COSMIC and MW TPW time series, respectively.

Figure 10. The global map of TPW trend over oceans using all F16, F17, Windsat data
from 2006 to 2013.

















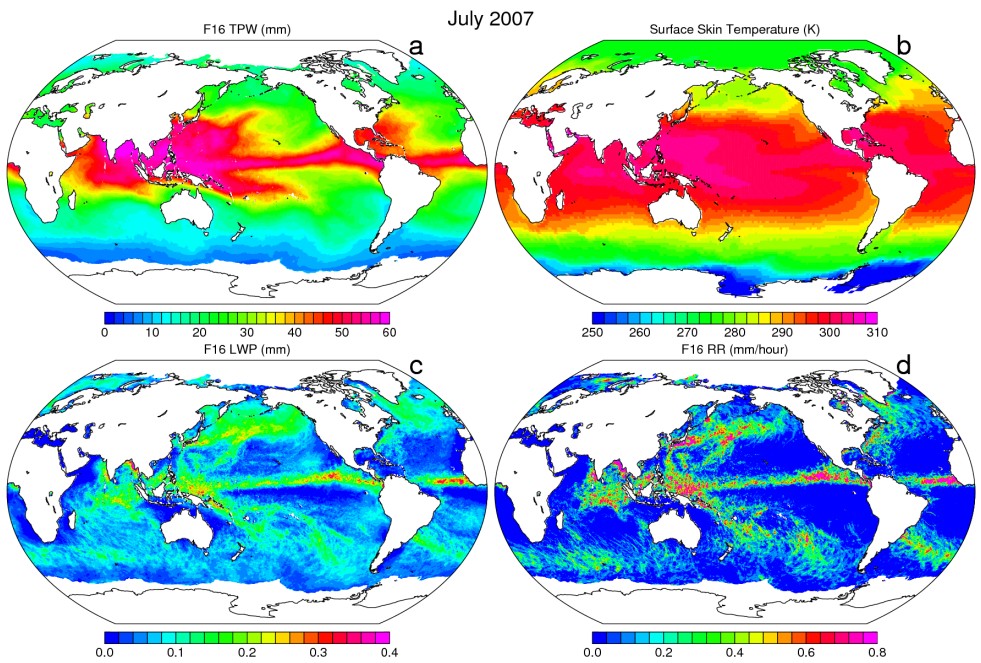

Figure 1. The RSS V7.0 monthly mean F16 SSM/I a) TPW (in mm), b) surface
skin temperature (in K), c) liquid water path (LWP, in mm), and d) rain rate (RR,
in mm/hour).





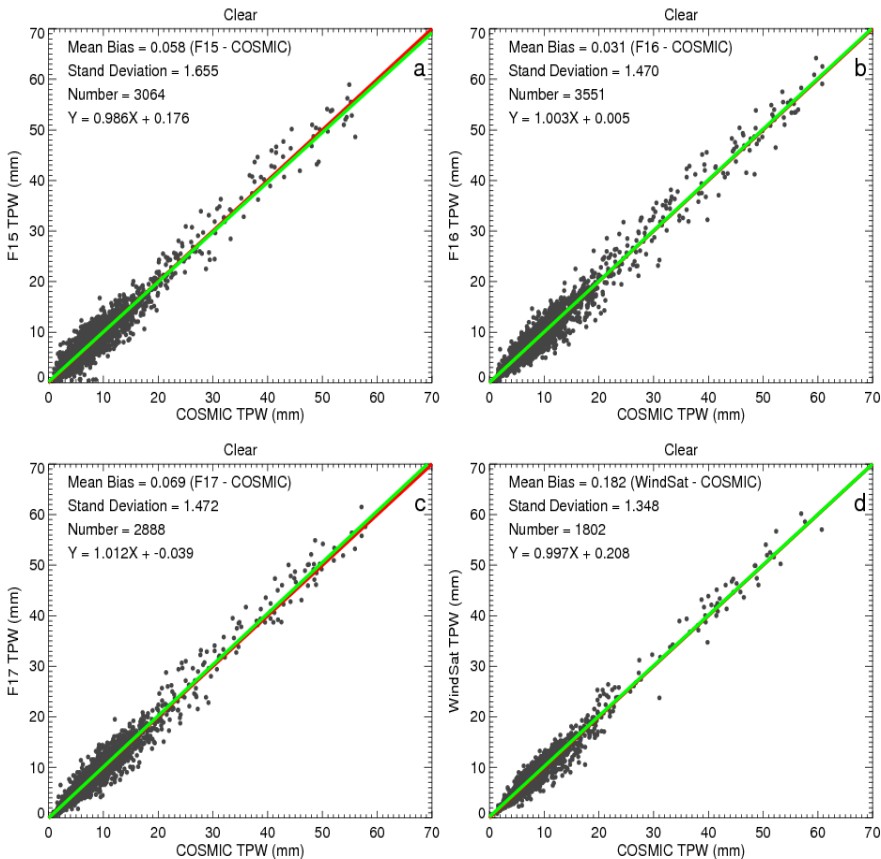

Figure 2. TPW scatter plots for the COSMIC and RSS Version 7.0 pairs under clear
condition for a) F15, b) F16, c) F17, and d) WindSat.

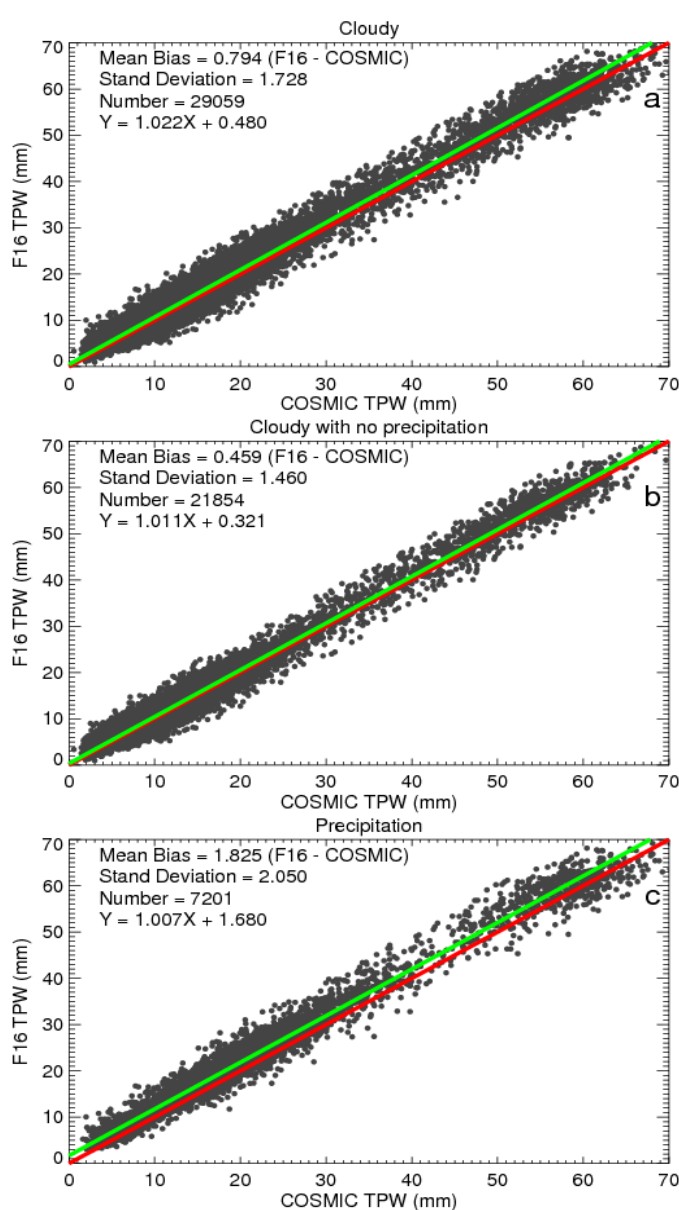

Figure 3. TPW scatter plots for the COSMIC and RSS Version 7.0 F16 SSM/I pairs
under a) cloudy, b) cloudy but non-precipitation, and c) precipitation conditions.




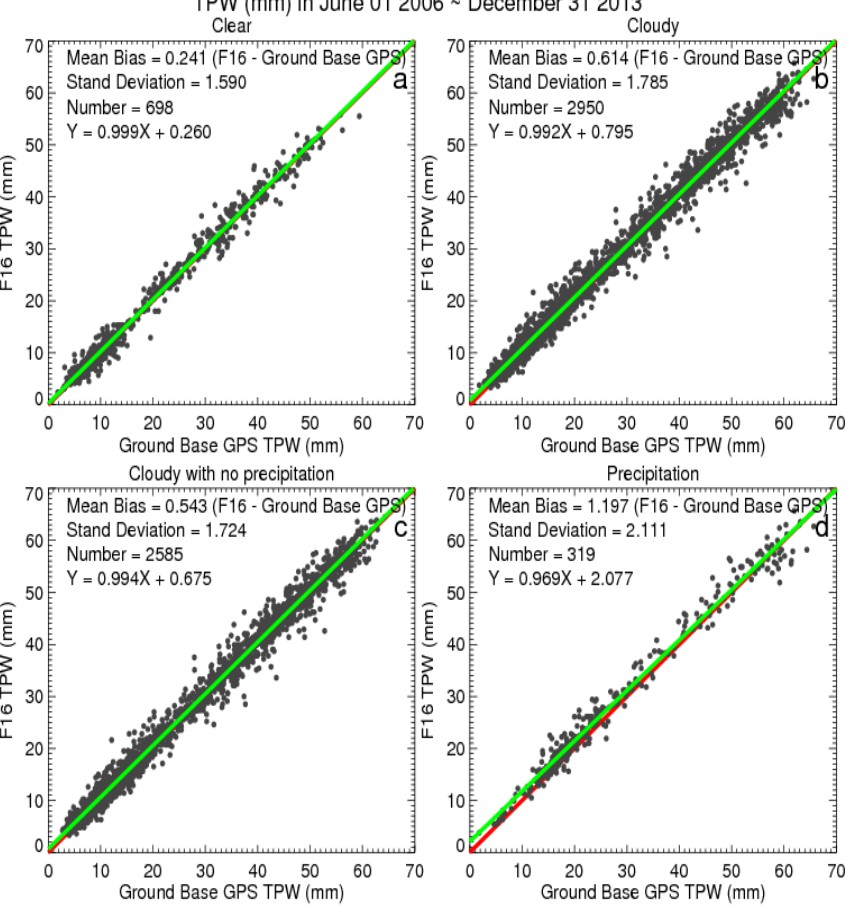

Figure 4. TPW scatter plots for the gb-GPS and RSS Version 7.0 F16 SSM/I pairs
from June 2006 to December 2013 under a) clear, b) cloudy, c) cloudy but non-
precipitation, and d) precipitation conditions.




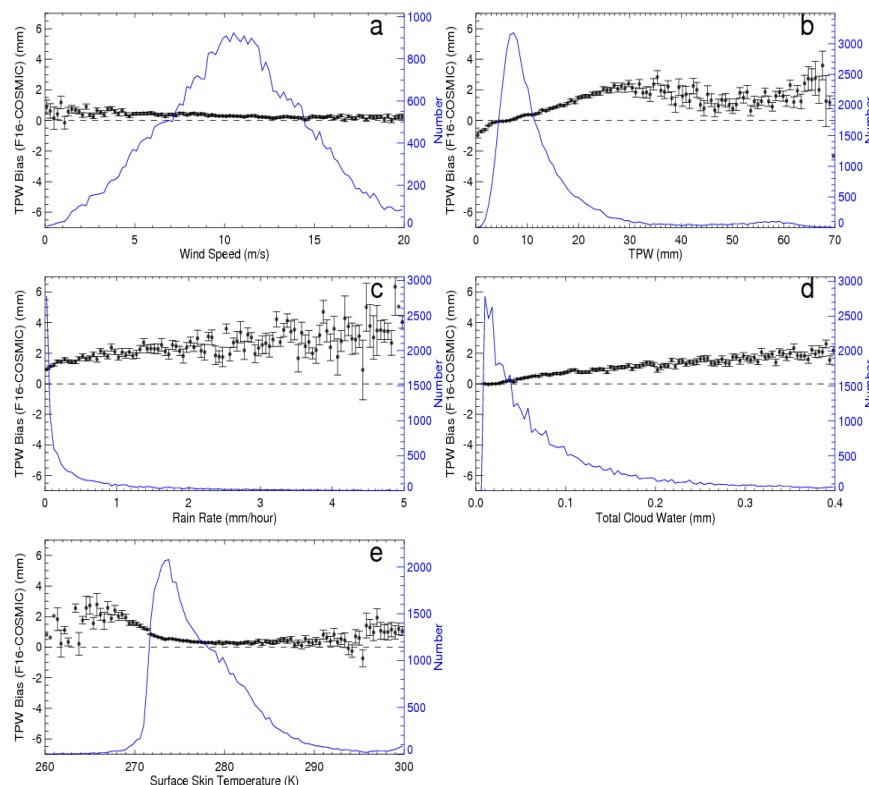

Figure 5. Mean and standard of the mean for the F16-COSMIC TPW biases varying with
a) wind speed (m/s), b) TPW (mm), c) rain rate (mm/hour), d) total cloud water (mm),
and e) surface skin temperature (K).  The vertical black bracket superimposed on the
mean denotes the standard error of the mean. The green dashed line is the number of
samples, indicated by the scale on the right.






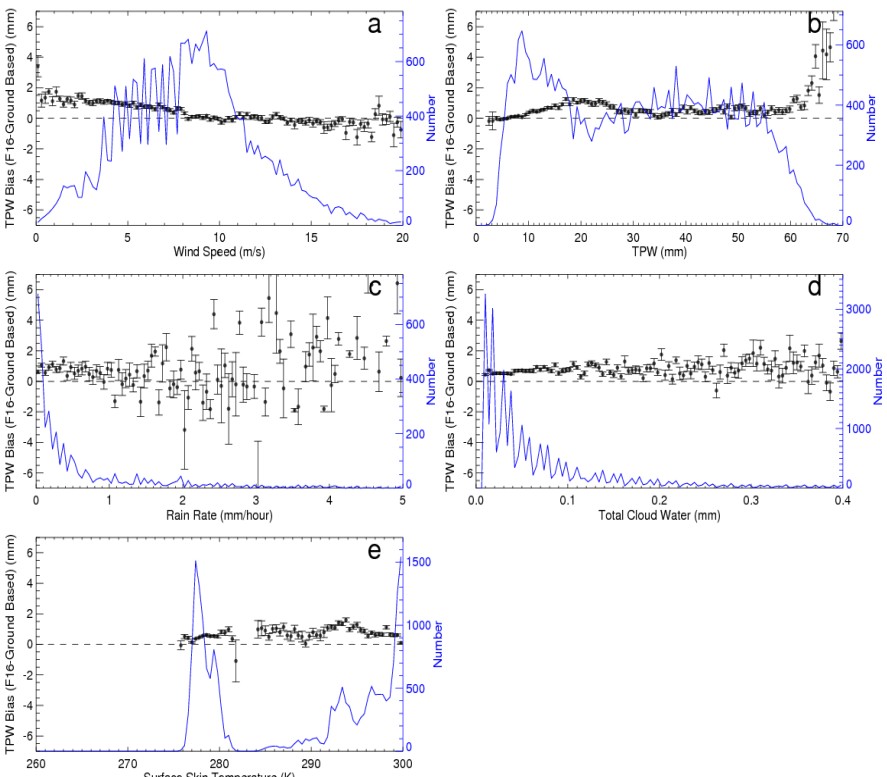

Figure 6. Mean and standard of the mean for the F16- gb-GPS TPW biases varying with
a) wind speed (m/s), b) TPW (mm), c) rain rate (mm/hour), d) total cloud water (mm) and
e) surface skin temperature (K).  The vertical black bracket superimposed on the mean
denotes the standard error of the mean. The green dashed line is the number of samples,
indicated by the scale on the right.




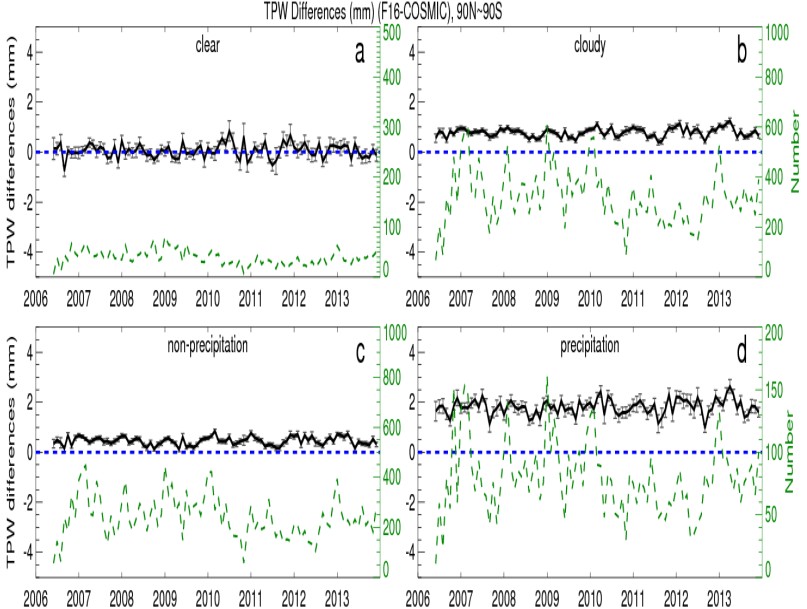

Figure 7. The time series of monthly mean F16 – COSMIC TPW differences under a)
clear, b) cloudy, c) cloudy but non-precipitation, and d) precipitation conditions. The
black line is the mean difference for microwave radiometer minus COSMIC; the vertical
lines superimposed on the mean values are the standard error of the mean. The number of
the monthly MW radiometer- COSMIC pairs is indicated by the green dashed line (scale
on the right Y axis).





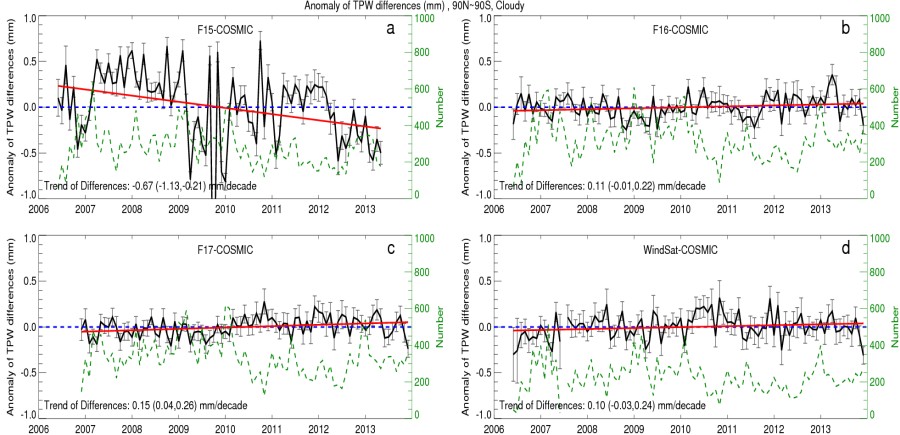

Figure 8. The time series of de-seasonalized TPW differences (microwave radiometer –
COSMIC) under cloudy skies for a) F15, b) F16, c) F17, d) WindSat. The black line is
the mean difference for microwave radiometer minus COSMIC; the vertical lines
superimposed on the mean values are the standard error of the mean. The number of the
monthly MW radiometer- COSMIC pairs is indicated by the green dashed line (scale on
the right Y axis). The trends are shown by solid red line. The 95% confidence intervals
for slopes are shown in the parentheses.

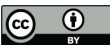


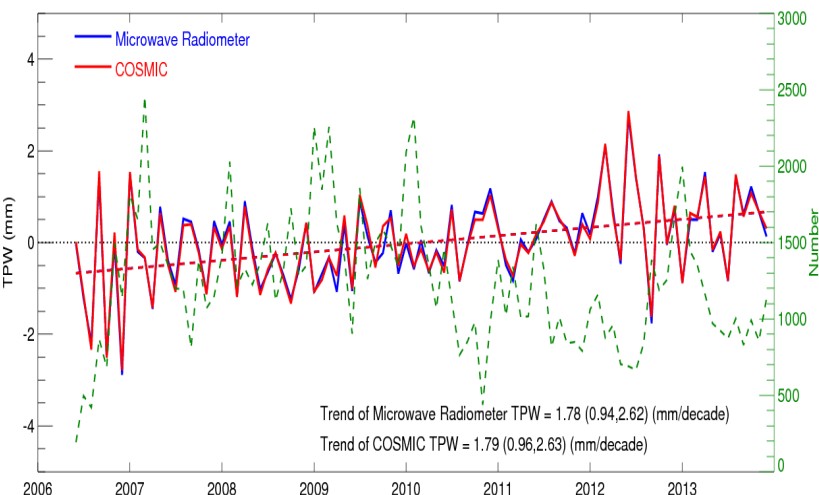

Figure 9. The de-seasonalized time series of global monthly mean TPW for all MW
instruments and COSMIC under all sky conditions. The red and blue dashed lines are the
best fit of de-seasonalized COSMIC and MW TPW time series, respectively.





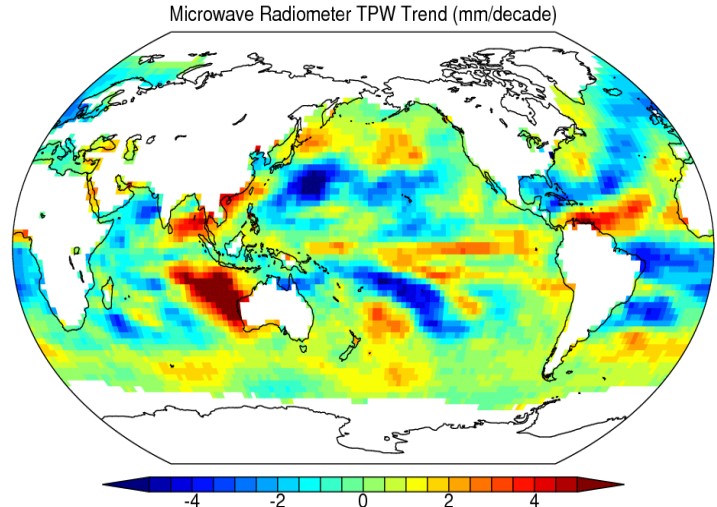

Figure 10. The global map of TPW trend over oceans using all F16, F17, Windsat
data from 2006 to 2013.