# Peer review of "1. Introduction"

_Atmospheric Chemistry and Physics, 2017_

## Referee Comment (RC1) · Anonymous Referee #1 · 12 Sep 2017

**Date: September 11, 2017**

Manuscript #:    acp-2017-525
Manuscript title: ***Comparison of Global Observations and Trends of Total Precipitable Water Derived from Microwave Radiometers and COSMIC Radio Occultation from 2006 to 2013***

**Brief Summary of the Manuscript**
This manuscript compares TPW estimates from the COSMIC radio occultation mission against estimates from SSM/I, SSMIS, radiometers, and WindSat over clear sky and cloudy conditions. The authors report a very good agreement between COSMIC and all other TWP data sets and trends. They also claim that the estimated differences between MW radiometers and COSMIC are mainly due to biases in the MW retrieval uncertainty under cloudy and precipitating conditions. This analysis is in my opinion a novel approach to establishing radio occultations as a new remote sensing climate instrument by cross-comparing the COSMIC results with independent data sets. The manuscript is very well written, coherent, and the results are presented nicely within the context of the investigation. My recommendation for this manuscript is publication after minor revisions, as described below.

**Major Comments:**
1) **Page 7; Line 147:** What is the cut-off value of the liquid water column? Given that this value establishes an upper limit in the estimation of the TPW in the RSS products, could this introduce a bias in the COSMIC vs RSS comparisons at high TPW values? I think this must be explicitly discussed in the manuscript.
2) **Page 11; Line 236:** The authors assume an 80% relative humidity below 0.1 km. What is the sensitivity of the COSMIC TPW estimation the relative humidity assumption? How does that affect the conclusions of this investigation?
3) **Lines 289–291, Lines 309–310, Lines 414-423:** The authors conclude that the primary source of the estimated biases between COSMIC and the rest of the data sets is the MW retrieval uncertainty. Because the largest biases are found under cloudy precipitating conditions, I think that the authors should also acknowledge that errors due to: a) cut-off liquid water and b) the 80% RH assumption below 0.1 km, could also contribute to the reported differences. Could there be a combined effect as well?
4) **Page 16; Line 357:** It should read: "...with F15, F16, F17, and WindSat under..."

**Minor Comments:**
a) **Line 57:** Grammatically the sentence is fine, but the noun "increases" reads rather awkward. Perhaps, consider replacing it with the word "enhancements"?
b) **Line 69:** It should read: "reanalyses".
c) **Line 92:** Place a comma after the word "Recently".
d) **Line 116:** Delete the word "and"
e) **Line 161:** It should read: "... (RR, in mm/hr), respectively, in 2007."
f) **Line 161:** It should read: "temperature variations over the Intertropical Convergence Zone (ITCZ) (Fig. 1b), which..."
g) **Lines 181-182:** It should read: "where P is the pressure in hPa, T is the temperature in K, $P_w$ is the water vapor pressure in hPa, $W_{water}$ is the liquid water content in grams per cubic meter (g m$^{-3}$)... "
h) **Line 208:** Place a comma after the word "troposphere".
i) **Lines 216-218:** I think that this statement is a bit bold. Perhaps, mention that the "...retrieved water vapor profiles are weekly dependent on the first guess" and provide a more appropriate reference that demonstrates that?
j) **Line 244:** Check "Wick2008". Is it written properly?
k) **Line 264:** Spell out December.
l) **Lines 264-266:** Delete this sentence. It appears twice in Lines 259-260.
m) **Line 268:** It should read: "Figures 2a-d..."
n) **Line 275:** Explain briefly how the "rad-cal" beacon biases the F15 data.

o) **Line 300:** It should read: "Figures 4a-d depict the..."
p) **Line 318:** It should read: "Figure 5b indicates that..."
q) **Line 321:** Delete "Fig. 5d"
r) **Line 323:** It should read: "Figure 5e shows that..."
s) **Lines 338, 342:** Spell out December.
t) **Line 353:** It should read: "Figure 8 depicts the..."
u) **Line 366:** It should read: "Figure 9 shows the..."
v) **Line 382:** It should read: "Figure 10 shows..."
w) **Line 383:** What about the F15 data?
x) **Line 385:** It should read: "...and west of Australia, south..."
y) **Lines 399–400:** It should read: "Because RO data have low sensitivity to clouds..."

---

## Referee Comment (RC2) · Anonymous Referee #2 · 17 Sep 2017

This study compares passive microwave (MW) estimates of total precipitable water (TPW) with radio occultation (RO) profiles of TPW that are closely matched together in space and time. The comparison is broken into four parts: clear sky, cloudy sky, cloudy sky with no precipitation, and cloudy sky with precipitation. The bias is smallest in clear sky and is largest within precipitating conditions. The bias is shown to be a small function of surface temperature, surface wind speed, etc., but these effects have little consequence on the interpretation of biases and trends, which lends further confidence to the results of this work. The trends in TPW are statistically significant

and are larger than previously reported. The trends are uniformly largest within cloudy non-precipitating skies, and can be slightly negative in clear sky for a few of the MW radiometers.

This is a very straightforward and useful study that is well written and flows logically. I only have a few minor comments and suggestions before this paper is accepted for publication.

Abstract and elsewhere: non-precipitating and precipitating conditions instead of non-precipitation and precipitation conditions? I'm not an expert in grammar but the latter reads a little odd.

Lines 52-54: is the global water vapor feedback still one of the largest uncertainties? We seem to know that the water vapor+lapse rate feedback has less spread in climate models than cloud feedbacks (see Soden et al., 2008, J. Climate, Figure 7, and other references). The role of water vapor and its regional variability, such as shown in Figure 10 in the manuscript, is probably the more uncertain quantity rather than global trends as shown in Figure 9. To summarize, it might be better to emphasize the role of water vapor in controlling cloud processes, and observing long term trends in water vapor is part of that understanding.

Line 66: land and ocean

Line 68: ocean

Line 195: IWC can be even a bit higher than that in convective towers, see D. Leroy et al., 2017, J. Atmos. Ocean Tech. that summarizes the HAIC/HIWC field campaign

Lines 233-234: what is the percent frequency of COSMIC water vapor profiles that sample below 0.1 km?

Lines 246-249: the larger spatial variance of water vapor in the tropics compared to the extratropics should be reflected in the higher standard deviations, and their increases sensitivity to collocation distance. Have the authors explored these differences? Would

also be helpful to cite a paper or two on the spatial variance of water vapor.

Line 250: a little bit of extra clarification on the matchups is warranted. Does one really get 20-60 MW pixels near a RO observation within a 1-hour period? This seems excessive. Is this at 0.25 degrees resolution or a larger distance? Are the matchups for the entire length of the 200 km RO or with respect to the tangent point at a particular reference altitude?

Line 268: Figures 3a-c (only three panels)

Line 298: under different

Line 310: a small but significant degree seems a little bit contradictory, maybe there is a better way to state this

Line 314: droplets

Line 385: Australian, and also South America

Line 412: reliable references

Lines 432-440: can the authors say anything about the magnitudes of these trends and whether they are consistent with the constant RH hypothesis of Earth's atmosphere?

Line 494: author list for reference is incomplete
* * *

---

## Author Comment (AC1) · 9 Nov 2017

Reviewer#1 comments

Manuscript #: acp-2017-525 Manuscript title: Comparison of Global Observations and Trends of Total Precipitable Water Derived from Microwave Radiometers and COSMIC Radio Occultation from 2006 to 2013

Brief Summary of the Manuscript:

[Figure]

This manuscript compares TPW estimates from the COSMIC radio occultation mission against estimates from SSM/I, SSMIS, radiometers, and WindSat over clear sky and cloudy conditions. The authors report a very good agreement between COSMIC and all other TWP data sets and trends. They also claim that the estimated differences between MW radiometers and COSMIC are mainly due to biases in the MW retrieval uncertainty under cloudy and precipitating conditions. This analysis is in my opinion a novel approach to establishing radio occultations as a new remote sensing climate instrument by cross-comparing the COSMIC results with independent data sets. The manuscript is very well written, coherent, and the results are presented nicely within the context of the investigation. My recommendation for this manuscript is publication after minor revisions, as described below.

=> We thank the reviewer for his or her thoughtful comments and have incorporated the suggested changes in the revised manuscript.

Major Comments:

1) Page 7; Line 147: What is the cut-off value of the liquid water column? Given that this value establishes an upper limit in the estimation of the TPW in the RSS products, could this introduce a bias in the COSMIC vs RSS comparisons at high TPW values? I think this must be explicitly discussed in the manuscript.

=> As shown in the valid data range reported by RSS (see http://www.remss.com/missions/ssmi/) the cut off values of the liquid water column are from -0.05 to 2.45 mm (plus the offset value -0.05 mm). => As demonstrated in Fig. 5d, the number of samples for RSS total cloud water (liquid water column) for those MW-COSMIC pairs peaks at around 0.01 mm ($\sim$2600) then decreases to fewer than 10 at 0.4 mm. The sample number for RSS total cloud water value equal to or larger than the cut-off value (2.40 mm) is therefore less than 10, which will not introduce any significant biases in the RSS MW-COSMIC comparison.

2) Page 11; Line 236: The authors assume an 80% relative humidity below 0.1 km.

[Figure]

What is the sensitivity of the COSMIC TPW estimation the relative humidity assumption? How does that affect the conclusions of this investigation?

=> We added the following in Section 2.3: The COSMIC TPW estimates are not very sensitive to the assumption of 80% relative humidity below 0.1 km (Step i above). The assumption of 80%+/-10% (i.e., 90% and 70%) relative humidity below 0.1 km introduces an uncertainty of about -/+0.03 mm uncertainty in the WV – COSMIC comparisons for all conditions. As shown in Section 4, this uncertainty is small compared to the observed differences between the RO and MW estimates.

3) Lines 289–291, Lines 309–310, Lines 414-423: The authors conclude that the primary source of the estimated biases between COSMIC and the rest of the data sets is the MW retrieval uncertainty. Because the largest biases are found under cloudy precipitating conditions, I think that the authors should also acknowledge that errors due to: a) cut-off liquid water and b) the 80% RH assumption below 0.1 km, could also contribute to the reported differences. Could there be a combined effect as well?

=> As stated in our responses to major comment 1), the RSS pre-defined cut off value for liquid water will not affect the conclusion from this study. => As stated in our responses to major comment 2), the 80% RH assumption below 0.1 km does not affect the conclusions from this study. => Since there is a very small number of RSS total cloud water values equal or larger than the cut-off value (2.40 mm), there is no combined effect for these two uncertainties that will affect the conclusion from this study.

4) Page 16; Line 357: It should read: "...with F15, F16, F17, and WindSat under..."

=> Done

Minor Comments:

a) Line 57: Grammatically the sentence is fine, but the noun "increases" reads rather awkward. Perhaps, consider replacing it with the word "enhancements"?

=> "increases" is replaced by "enhancements"

b) Line 69: It should read: "reanalyses".

=> All of the "reanalysis" are replaced by "reanalyses".

c) Line 92: Place a comma after the word "Recently".

ïč£ Done

d) Line 116: Delete the word "and"

=> Done

e) Line 161: It should read: "... (RR, in mm/hr), respectively, in 2007."

=> Done

f) Line 161: It should read: "temperature variations over the Intertropical Convergence Zone (ITCZ) (Fig. 1b), which..."

=> Done

g) Lines 181-182: It should read: "where P is the pressure in hPa, T is the temperature in K, Pw is the water vapor pressure in hPa, Wwater is the liquid water content in grams per cubic meter (g m-3)... "

=> Done

h) Line 208: Place a comma after the word "troposphere".

=> Done

i) Lines 216-218: I think that this statement is a bit bold. Perhaps, mention that the "...retrieved water vapor profiles are weekly dependent on the first guess" and provide a more appropriate reference that demonstrates that?

=> In Line 226, "the retrieved water vapor profiles are insensitive to the first guess water vapor profiles" is replaced with "the retrieved water vapor profiles are weakly dependent on the first guess water vapor profiles (Neiman et al. 2008)". Neiman et al.

(2008) is a good reference for this statement.

j) Line 244: Check "Wick2008". Is it written properly?

=> In Line 125 of the original manuscript (141-142 of revised manuscript), we defined Wick et al.(2008) as "Wick2008" the first time the reference is given, so "Wick2008" is ok.

k) Line 264: Spell out December.

=> Done

l) Lines 264-266: Delete this sentence. It appears twice in Lines 259-260.

=> Done

m) Line 268: It should read: "Figures 2a-d..."

=> Done

n) Line 275: Explain briefly how the "rad-cal" beacon biases the F15 data.

=> One sentence is added in Line 285, "On 14 August 2006, a radar calibration beacon (RAD-CAL) was activated on F15. This radar interfered with the SSM/I, primarily the 22V channel, which is a key channel for water vapor retrievals. Although a correction method derived by Hilburn and Wentz (2008) and Hilburn (2009) was applied, the 22 V channel is not being full corrected (Wentz, 2012). As a result, there are still errors in the water vapor retrievals." Âă

=> Two papers are added in to references:

Hilburn, K. A. and F. J. Wentz, 2008: Mitigating the Impact of RADCAL Beacon Contamination on F15 SSM/I Ocean Retrievals. Geophysical Research Letters, 35. L18806, doi:10.1029/2008GL034914.Âă

Hilburn, K. A., 2009: Including Temperature Effects in the F15 RADCAL Correction. RSS Technical Report 051209, Remote Sensing Systems, Santa Rosa, CA,

http://www.remss.com/papers/RSS_TR051209_RADCAL.pdf.

o) Line 300: It should read: "Figures 4a-d depict the..."

=> Done

p) Line 318: It should read: "Figure 5b indicates that..."

=> Done. And all "Fig." replaced with "Figure" in the paper

q) Line 321: Delete "Fig. 5d" => Done

r) Line 323: It should read: "Figure 5e shows that..."

=> Done

s) Lines 338, 342: Spell out December.

=> Done

t) Line 353: It should read: "Figure 8 depicts the..."

=> Done

u) Line 366: It should read: "Figure 9 shows the..."

=> Done

v) Line 382: It should read: "Figure 10 shows..."

=> Done

w) Line 383: What about the F15 data?

=> The reason we did not include F15 data in Figure 10 is mentioned in the last para on page 17 of the revised manuscript: "The reason for larger standard deviations of the MW minus RO differences for F15 (Tables 2 and 3 and Fig. 8a) is very likely because the F15 data after August 2006 were corrupted by the "rad-cal" beacon that was turned on at this time." Also on this page "RSS does not recommend using these

measurements for studies of long-term change. Thus, we consider the F15 data less reliable during the period of our study."

x) Line 385: It should read: "...and west of Australia, south..."

=> Done

y) Lines 399–400: It should read: "Because RO data have low sensitivity to clouds..."

=> Done

---

## Author Comment (AC2) · 9 Nov 2017

This study compares passive microwave (MW) estimates of total precipitable water (TPW) with radio occultation (RO) profiles of TPW that are closely matched together in space and time. The comparison is broken into four parts: clear sky, cloudy sky, cloudy sky with no precipitation, and cloudy sky with precipitation. The bias is smallest in clear sky and is largest within precipitating conditions. The bias is shown to be

a small function of surface temperature, surface wind speed, etc., but these effects have little consequence on the interpretation of biases and trends, which lends further confidence to the results of this work. The trends in TPW are statistically significant and are larger than previously reported. The trends are uniformly largest within cloudy non-precipitating skies, and can be slightly negative in clear sky for a few of the MW radiometers.

This is a very straightforward and useful study that is well written and flows logically. I only have a few minor comments and suggestions before this paper is accepted for publication.

=> We thank this reviewer for his or her thoughtful comments, and we have incorporated them into the revised manuscript.

1. Abstract and elsewhere: non-precipitating and precipitating conditions instead of non-precipitation and precipitation conditions? I'm not an expert in grammar but the latter reads a little odd.

=> To be consistent, we use precipitating and non-precipitating throughout in the revised paper.

2. Lines 52-54: is the global water vapor feedback still one of the largest uncertainties? We seem to know that the water vapor+lapse rate feedback has less spread in climate models than cloud feedbacks (see Soden et al., 2008, J. Climate, Figure 7, and other references). The role of water vapor and its regional variability, such as shown in Figure 10 in the manuscript, is probably the more uncertain quantity rather than global trends as shown in Figure 9. To summarize, it might be better to emphasize the role of water vapor in controlling cloud processes, and observing long term trends in water vapor is part of that understanding.

=> We have revised the introductory paragraph to incorporate these comments. In addition, we added the following paper to the references:

Held, I. M., and B. J. Soden, 2000: Water vapor feedback and global warming, Annu. Rev. Energy Environ., 25, 441–475, doi:10.1146/annurev.energy.25.1.441.

Soden, B.J. and I.M. Held, 2006: Assessment of climate feedbacks in coupled ocean-atmosphere models. J. Climate, 19, 3354-3360.

Wentz, F.J., Lucrezia Riccardulli, K. Hilburn, and C. Mears, 2007: How much more rain will global warming bring? Science, 317, 233-235.

3. Line 66: land and ocean => We think it is more proper to use "lands and oceans". => We changed this to "land areas and oceans.

4. Line 68: ocean

=> We still use "oceans" since this means all "oceans".

5. Line 195: IWC can be even a bit higher than that in convective towers, see D. Leroy et al., 2017, J. Atmos. Ocean Tech. that summarizes the HAIC/HIWC field campaign

=> We added "Heymsfield et al., (2002) reported high ice water content values ranging from 0.1 – 0.5 gm-3 in tropical cirrus and stratiform precipitating clouds, although it may rarely reach as high as 1.5 gm-3 in deep tropical convective clouds (Leroy et al., 2017)." We also added two references:

Leroy, D., Fontaine, E., Schwarzenboeck, A., Strapp, J. W., Korolev, A., McFarquhar, G., Dupuy, R., Gourbeyre, C., Lilie, L., Protat, A., Delanoë, J., Dezitter, F., and Grandin, A., 2017: Ice crystal sizes in high ice water content clouds. Part 2: Statistics of mass diameter percentiles in tropical convection observed during the HAIC/HIWC project, J. Atmos. Oceanic Technol., doi: 10.1175/JTECH-D-15-0246.1.

Heymsfield, A. J., A. Bansemer, P. R. Field, S. L. Durden, J. L. Stith, J. E. Dye, W. Hall, and C. A. Grainger, 2002: Observations and parameterizations of particle size distributions in deep tropical cirrus and stratiform precipitating clouds: Results from in situ observations in TRMM field campaigns. J. Atmos. Sci., 59, 3457–3491,

doi:10.1175/1520-0469(2002)059, 3457.

6. Lines 233-234: what is the percent frequency of COSMIC water vapor profiles that sample below 0.1 km?

=> We added "About 70% to 90% of COSMIC profiles reach to within 1 km of the surface (Anthes et al, 2008). Usually more than 30% of COSMIC water vapor profiles reach below 0.1 km in the mid-latitudes and higher latitudes, and a little bit less than 10% in the tropical regions."

7. Lines 246-249: the larger spatial variance of water vapor in the tropics compared to the extratropics should be reflected in the higher standard deviations, and their increases sensitivity to collocation distance. Have the authors explored these differences? Would also be helpful to cite a paper or two on the spatial variance of water vapor.

=> Yes, the larger spatial variance of water vapor in the tropics compared to the extratropics (see Fig. 1a) should be reflected in the higher standard deviations. However, because RSS TPW retrieval errors could also be larger over stronger convective regions under cloud and precipitating conditions (see section 4 and Figs. 5 and 6), it is hard to distinguish the tropical vs. sub-tropical effect from retrieval uncertainty effects.

=> We added two sentences in the end of the paragraph "Note that, although not shown, the mean biases and standard deviations of the mean biases are slightly larger over the tropics than those over mid-latitudes. This could be because of the combined effect of the larger spatial TPW variation in the tropics than those in the mid-latitudes (see Fig. 1a, and Neiman et al., 2008; Teng et al., 2013; Mears et al., 2015) and the fact that the MW TPW retrieval uncertainty is also larger over stronger convection regions. More results are detailed in Section 4."

8. Line 250: a little bit of extra clarification on the matchups is warranted. Does one really get 20-60 MW pixels near a RO observation within a 1-hour period? This seems

excessive. Is this at 0.25 degrees resolution or a larger distance? Are the matchups for the entire length of the 200 km RO or with respect to the tangent point at a particular reference altitude?

=> As mentioned in Section 2.3, "Pairs of MW and RO TPW estimates collocated within 50 km and one hour are collected." Over tropical regions, 0.25 degrees is about 25 km. So one RO observation can match with about 16 (4X4) 0.25x0.25 MW grids. In higher latitudes, a 0.25 degree resolution is less than 25 km. So within 50 km radius (100 km diameter), one RO observation can match from 16 (4x4) to 49 (7x7 in very high latitude) MW 0.25x0.25 grids. Although it is not mentioned in the text, there are about 1 to 2 MW pixels binned into each 0.25X0.25 grid. Therefore, in the text, we mention that we will have about 20-60 MW pixels near a RO observation.

=> Again, the matchup is within 50 km at the location of RO tangent point at 4-5 km altitude. We added the statement "The location of RO observation is defined by the RO tangent point at 4-5 km altitude."

9. Line 268: Figures 3a-c (only three panels)

=> It should refer to Fig. 2 not Fig. 3, and is revised to "Figures 2a-d"

10. Line 298: under different

=> Done

11. Line 310: a small but significant degree seems a little bit contradictory, maybe there is a better way to state this

=> "a small but significant degree" is replaced by "which in turn affect the MW TPW retrievals.

12. Line 314: droplets

=> Done

13. Line 385: Australian, and also South America

=> Done

14. Line 412: reliable references

=> "reliable reference data" is used.

15. Lines 432-440: can the authors say anything about the magnitudes of these trends and whether they are consistent with the constant RH hypothesis of Earth's atmosphere?

=> This is a good question, but also one that is difficult to answer from our results. We added the following to the end of the Discussion section:

Other studies have suggested that this positive feedback results in a nearly constant global mean relative humidity (Soden and Held, 2006; Sherwood et al., 2010). However, it is difficult to directly relate our estimated TPW trends to constant RH hypothesis of Earth's atmosphere under global warming. The global mean surface temperature has been rising at about the rate of 0.2 K/decade in the past twenty years. A 0.2K increase in temperature would produce about a 1.4% increase in saturation water vapor pressure based on the Clausius-Clapeyon equation. To maintain a constant RH for this temperature increase, the actual water vapor pressure (and specific humidity) would also have to increase by 1.4%. In this study, we observe an increase of TPW in our dataset of about 1.78 mm/decade which is 6.9 percent increase per decade in TPW. Our dataset is dominated mainly by cloudy samples over middle latitudes (40N-60N and 40-65S). Thus, from these numbers alone we would expect an increase in mean RH under cloudy conditions by more than 6%, which is unlikely and well outside the range of changes in relative humidity in models (e.g. Figure 2 in Sherwood et al., 2010). However, the changes in the global mean RH are not related in such a simple fashion to changes in the global mean temperature and precipitable water. For example, Figure 10 depicts that there are very large differences in the spatial distribution of TPW

changes, which shows regional variations of +/- 4 mm/decade. Thus, some regions are drying and others are moistening. The variations in global mean surface temperature are also large, but very different from those of TPW, with the polar regions and continents warming up much faster than the atmosphere over the oceans. In cold polar regions, an increase in temperature will result in a smaller increase in saturation vapor pressure than the same increase in temperature in the tropics. The global evaporation and precipitation patterns also vary greatly, as water vapor transport is important in the global water vapor balance. All of this, as discussed by Held and Soden (2000), Soden and Held (2006), and Sherwood et al. (2010) means that the relationships between global mean temperature increase, TPW changes, and the resulting change in global mean RH are not simple.

16. Line 494: author list for reference is incomplete

=> All of the authors are added in the reference. The new reference is "Fetzer, E. J., W.G. Read, D. Waliser, B. H. Kahn, B. Tian, H. Vömel, F. W. Irion, H. Su, A. Eldering, M. T. Juarez, J. Jiang, and V. Dang, 2008: Comparison of upper tropospheric water vapor observations from the Microwave Limb Sounder and Atmospheric Infrared Sounder. J. Geophys. Res., 113/D22, D22110."